# Biological Function and Application of Picornaviral 2B Protein: A New Target for Antiviral Drug Development

**DOI:** 10.3390/v11060510

**Published:** 2019-06-04

**Authors:** Zengbin Li, Zixiao Zou, Zeju Jiang, Xiaotian Huang, Qiong Liu

**Affiliations:** 1School of Public Health, Nanchang University, Nanchang 330006, China; nculizengbin@163.com; 2Department of Medical Microbiology, School of Medicine, Nanchang University, Nanchang 330006, China; zzx1019446427@163.com (Z.Z.); xthuang@ncu.edu.cn (X.H.); 3Jiangxi Medical College, Nanchang University, Nanchang 330006, China; jiangzeju1998@126.com

**Keywords:** picornavirus, 2B protein, viroporin, drug development

## Abstract

Picornaviruses are associated with acute and chronic diseases. The clinical manifestations of infections are often mild, but infections may also lead to respiratory symptoms, gastroenteritis, myocarditis, meningitis, hepatitis, and poliomyelitis, with serious impacts on human health and economic losses in animal husbandry. Thus far, research on picornaviruses has mainly focused on structural proteins such as VP1, whereas the non-structural protein 2B, which plays vital roles in the life cycle of the viruses and exhibits a viroporin or viroporin-like activity, has been overlooked. Viroporins are viral proteins containing at least one amphipathic α-helical structure, which oligomerizes to form transmembrane hydrophilic pores. In this review, we mainly summarize recent research data on the viroporin or viroporin-like activity of 2B proteins, which affects the biological function of the membrane, regulates cell death, and affects the host immune response. Considering these mechanisms, the potential application of the 2B protein as a candidate target for antiviral drug development is discussed, along with research challenges and prospects toward realizing a novel treatment strategy for picornavirus infections.

## 1. Introduction

The *Picornaviridae* family consists of 35 genera and 80 species, mainly including *Enterovirus*, *Hepatovirus*, *Cardiovirus*, *Aphthovirus*, *and Rhinovirus* [1]. To date, research on picornaviruses has mainly focused on enterovirus (EV) 71, coxsackievirus (CV), poliovirus (PV), encephalomyocarditis virus (EMCV), foot-and-mouth disease virus (FMDV), human rhinovirus (HRV), and hepatitis A virus (HAV). Picornavirus infections can cause enormous damage in humans and animals. The EV71, CVA16, and CVA10 cause hand, foot, and mouth disease in millions of children in Asia-Pacific region each year and can cause more serious clinical symptoms such as aseptic meningitis, acute flaccid paralysis, and neurological respiratory syndrome [1,2,3]. 

Picornaviruses are non-enveloped spherical viruses with an icosahedral-structured viral capsid. The picornaviral genome consists of a single-stranded positive-sense RNA, which is approximately 6.7–10.1 kilobases in length, with a highly conserved structure, including a 5′-noncoding region (5’-NCR), an open reading frame, a 3′-NCR, and a 3′-end polyA tail [1]. The 5′-NCR contains multiple RNA secondary structural elements, including the internal ribosome entry site. The open reading frame of the viral genome consists of three regions: P1, P2, and P3. The P1 region is translated and processed to form the structural proteins VP1, VP2, VP3, and VP4, which compose the capsid structure of a picornavirus. The P2 and P3 regions are separately translated to the non-structural proteins 2A, 2B, and 2C and 3A, 3B, 3C, and 3D, respectively. The majority of related research has focused on structural proteins of picornaviruses, such as VP1, whereas the importance of the non-structural protein 2B has been relatively overlooked. 

Viroporins are proteins found in a variety of viruses and are generally comprised of 50 to 120 amino acids. Each viroporin contains a highly hydrophobic domain capable of forming at least one amphipathic α-helical structure, which oligomerizes to form transmembrane hydrophilic pores [4,5]. The 2B protein is a crucial component of picornaviruses that exhibits viroporin or viroporin-like activity, plays a key role in the picornavirus life cycle by inducing a series of cytotoxic reactions to promote picornaviral replication and release [6,7,8,9,10,11,12,13,14]. The 2B protein has a highly conserved sequence, which can be exploited for viral detection [15,16,17], vaccine development [18,19,20], and RNA interference [21,22,23,24,25]. In addition, the 2B protein exhibits a viroporin or viroporin-like activity, and thus, targeted drugs against viroporin could potentially target 2B protein as a novel strategy to treat or prevent picornavirus infections. However, the detailed mechanism of action of the 2B protein has not been elucidated to date. Therefore, here, we review the recent research data on the role of the 2B protein in the picornaviral life cycle and discuss its possible application in antiviral therapy. 

## 2. Structure and Cellular Location of the Picornaviral 2B Protein

The picornaviral 2B protein is a relatively short molecule, containing a maximum of two predicted putative transmembrane hydrophobic helices, along with N- and C-terminal domains, which are connected by a short stretch of amino acid residues. The α-helix-turn-α-helix sequence of the 2B protein is the basis for forming a transmembrane pore through homo-multimerization and the major determinant of the 2B protein function [4,11,26,27,28,29,30]. A computational approach has demonstrated that the EV 2B protein is a tetramer, and 2B proteins with different orientations have different activities [31,32]. The 2B protein belongs to the type II family of viroporins, which can be further divided into different types according to the number and orientation of the membrane-spanning domains (Figure 1). In type IIA viroporins, the N- and C-termini stretch to the organelle lumen, such as in the 2B protein of CVB3 [6,9,26,33], whereas the N- and C-termini of type IIB viroporins face the cytoplasmic matrix, such as in PV [9,27,34] and FMDV [7,14,28]. In addition, the C-terminus of the HAV 2B protein has a viroporin-like activity [11]. 

Picornaviral 2B proteins target the membrane and form pores mainly through their transmembrane regions. The protein molecules are first inserted into the membrane individually and then self-interact and homo-oligomerize to form higher-order structures, which are important for the pore-forming activity, determined by the specific sequence and structure [4,34,35]. The majority of 2B proteins are localized to organelles, with predominant co-localization with the Golgi apparatus and the endoplasmic reticulum (ER) in CVB3, PV, and HRV14 (Figure 1) [36]. The HRV16 and FMDV 2B proteins are mainly localized to the ER, whereas the EMCV 2B protein is not localized to either the Golgi complex or the ER [28,36,37]. Seggewiss et al. [38] found that the HAV 2B protein was not localized to the ER either but was involved in the amendment of the ER–Golgi apparatus intermediate compartment. Since the protein structure determines its ultimate function and 2B proteins belonging to different viroporin species show both similarities and differences in their functions, much insight can be gained from research on the same viroporin 2B protein and on different viroporins.

## 3. Biological Functions of the Picornaviral 2B Protein

The 2B protein can induce many cellular reactions, such as changing membrane permeability, regulating apoptosis and autophagy, and affecting host immune responses. These functions are all related to changes in ion concentrations, especially of calcium ions (Ca^2+^). Therefore, here, we mostly focus on the role of Ca^2+^ in these 2B protein activities. 

### 3.1. Change in Membrane Permeability

A common feature of infection by animal viruses is the damage to the ion balance in host cells. The picornaviral 2B protein may change the membrane permeability of target cells, disturbing the ion balance, especially that of Ca^2+^, in organelles, such as the ER and the Golgi apparatus (Figure 1) [6,36,39]. The changes in membrane permeability, caused by the 2B protein, have also been suggested to be regulated by the content of specific membrane phospholipids [11,40]. The Ca^2+^ are involved in the activation of enzymes in cells and play a crucial role in viral replication and other viral biological processes [41,42,43]. However, the role of the 2B protein in Ca^2+^ homeostasis remains unclear.

Initial studies only indicated that host cells had elevated the Ca^2+^ levels, owing to the expression of the 2B protein [9,28,29,36,44,45,46], but the mechanism was not clarified. Since then, some researchers have proposed that the decrease in the concentration of Ca^2+^ stored in organelles triggers the opening of specific calcium ion channels on the plasma membrane of cells, causing an influx of extracellular Ca^2+^ [29]. This idea was supported by the findings that expression of the CVB3 and PV 2B proteins resulted in an obvious decrease in the Ca^2+^ concentration in the ER and Golgi complex, along with a decrease in calcium uptake by the mitochondria. Meanwhile, the increased Ca^2+^ level in the cytoplasm was suggested to be mainly due to the influx of extracellular Ca^2+^ [36,44,45]. Similarly, the expression of the HRV 2B protein was shown to decrease the Ca^2+^ concentrations in the ER and Golgi apparatus, whereas the EMCV 2B protein only significantly reduced the Ca^2+^ concentration in the ER [36]. In contrast, other studies showed that expression of the HAV and FMDV 2B proteins elevated the cytoplasmic Ca^2+^ level but did not alter the level of stored Ca^2+^ in organelles, such as the ER and Golgi complex [28,36]. Taken together, these studies suggest that there are different mechanisms by which 2B proteins affect the Ca^2+^ concentrations, depending on the virus type. Furthermore, it is unknown whether Ca^2+^ directly pass through the channel formed by the 2B protein. Pham et al. [47] demonstrated, using a planar lipid bilayer and liposome patch-clamp electrophysiological technique, that the rotavirus non-structural protein 4 (NSP4) viroporin region acts as a Ca^2+^ conduction channel. 

Although there is currently no direct evidence that the 2B protein can directly induce the observed changes in the Ca^2+^ concentration in host cells upon infection, the above-reviewed studies suggest an association, and the mechanism requires further investigation. Given the importance of Ca^2+^ signaling for numerous cellular processes, further studies on picornaviral 2B protein function should include determination of the Ca^2+^ concentration, which may provide more insight into the detailed function of the 2B protein. In particular, the 2B protein may change the Ca^2+^ concentration to regulate autophagy and apoptosiswhich are distinct cell death mechanisms controlled by the virus to effectively evade the host immunity, thereby promoting viral replication and release [48,49,50,51,52]. 

### 3.2. Regulation of Host Cell Apoptosis and Autophagy

Picornaviruses can form new cytoplasmic vesicles by inducing membrane remodeling, thereby promoting their own proliferation [39,53,54]. The 2B protein is capable of binding to the membrane and inducing target membrane remodeling to form a unique membrane structure that can serve as a viral replication site. This site, known as the viroplasm, is generated from the ER to accumulate all of the cellular components required for viral replication (Figure 2) [13,39,54,55,56]. The viroplasm is also the main membrane source of autophagy [54,57]. 

The CVB3 2B protein is dependent on its transmembrane hydrophobic region to induce autophagy [8], which may be related to alterations in membrane permeability, especially with regard to the Ca^2+^ concentration. Moreover, at an early stage of FMDV cell infection, the virus specifically recognizes and binds to the cell surface receptors, and the 2B protein rapidly upregulates the autophagy pathway, leading to punctate aggregation of a large number of autophagy marker proteins, such as the microtubule-associated protein 1 light chain 3 (MAP1-LC3) [28,58]. In addition, rotavirus encodes the NSP4 viroporin, which releases the ER-stored Ca^2+^ into the cytoplasm, thereby activating the Ca^2+^/calmodulin-dependent kinase kinase-β (CaMKK-β) signaling pathway, leading to autophagy (Figure 2) [59]. Further, CVB4 induces autophagy in a calpain-dependent manner, causing an accumulation of LC3 lipids and autophagosomes [60]. Considering the ability of the 2B protein to alter cellular calcium homeostasis, along with its viroporin-like activity, it is feasible that the 2B protein may regulate autophagy mainly by changing the Ca^2+^ concentration.

The 2B protein has also been shown to regulate apoptosis through the endogenous pathway, which can be divided into ER stress and the mitochondrial pathway, providing another potential mechanism of bypassing the host immune response to facilitate infection [32,37,44,45,48]. The Ca^2+^ plays a pivotal role in ER stress-dependent apoptosis by regulating the flow between the ER and the mitochondria [45,61]. Excessive mitochondrial uptake of Ca^2+^ exerts a cytotoxic effect because a high Ca^2+^ concentration can open numerous mitochondrial transition pores, increase mitochondrial permeability, and destroy the mitochondrial outer membrane; consequently, cytochrome c and other proapoptotic factors are released, leading to apoptosis (Figure 2) [9,32,44,45,62]. The CVB3 2B protein was shown to inhibit caspase activation and cell death induced by actinomycin D and cycloheximide by regulating the intracellular Ca^2+^ concentration [44,45]. Additionally, the 2B protein of HRV16 induced an ER stress response, accompanied by an increased expression of cleaved caspase-3 and CCAAT-enhancer-binding protein homologous protein (CHOP), which might have also involved a change in the Ca^2+^ level [37]. Collectively, these results suggest that the 2B protein may regulate apoptosis by altering calcium homeostasis. Furthermore, the 2B protein can regulate apoptosis through the mitochondrial pathway. Madan et al. [32] showed that the PV 2B protein interacted with the mitochondria and altered the mitochondrial morphology, in addition to the release of cytochrome c after the PV *2B* gene expression. Cong et al. [63] reported that the EV71 2B protein was localized to the mitochondria and induced apoptosis by directly activating the proapoptotic B-cell lymphoma 2-associated X (BAX) protein, without a significant uptake of Ca^2+^ by the mitochondria. Therefore, the activation of the mitochondrial apoptotic pathway and subsequent apoptosis, induced by the EV71 2B protein, may not involve Ca^2+^ signaling. Collectively, picornaviral 2B proteins can induce cell death in a variety of ways, with Ca^2+^ playing an important role in most of these mechanisms.

### 3.3. Effect on the Host Immune Response

The host immune system is an important line of defense against pathogens, and pathogens can affect the immune system in a variety of ways. The 2B protein mainly affects the host immune response through inflammasome activation and by direct antagonism of the host immune response. Recognition of pathogens by the immune system is mainly mediated by pathogen-associated molecular pattern receptors, known as pattern recognition receptors, including nucleotide-binding oligomerization domain (NOD)-like receptors (NLRs), retinoic acid-inducible gene-I (RIG-I)-like helicases, and pyrin domain-containing 3 (NLRP3) [64,65,66]. 

Activation of NLRP3 inflammasome occurs during a period of changes in ion concentrations [67,68]. The NLRP3 belongs to the NLR family of inflammasomes and causes interleukin (IL)-1β and IL-18 secretion via caspase-1 activation [67]. The EMCV, PV, EV71, and HRV 2B proteins all activate the NLRP3 inflammasome but use distinct mechanisms [12,69]. The HRV and EMCV 2B proteins can stimulate the NLRP3 inflammasome pathway to activate caspase-1, which catalyzes the proteolysis of pro-IL-1β to IL-1β, leading to its secretion from across the plasma membrane by inducing a Ca^2+^ efflux from intracellular storage (Figure 3) [12,69]. Wang et al. [70] found that CVB3-infected cells induced the NLRP3 activation in association with a K^+^ efflux. The influenza virus M2 protein, which is also a viroporin, is capable of transporting Na^+^ and K^+^, resulting in activation of the NLRP3 inflammasome [71,72]. Since the CVB3 2B protein acts as a viroporin and can disrupt the intracellular ion balance [36,46], it has been speculated that the induction of NLRP3 activation in CVB3-infected cells may be related to the 2B protein.

In addition to activating the inflammasome, the 2B protein also antagonizes the host immune response. Both in vitro and in vivo studies have suggested that inhibition of protein trafficking would effectively allow viral evasion of the host immune response (Figure 3) [73,74,75]. Moreover, inhibition of protein transport may be related to changes in the Ca^2+^ concentration [46]. Similar to the CVB3 2B protein, the 2B proteins of PV, HRV16 and HRV14, were shown to significantly inhibit the protein transport through the Golgi complex, whereas the HAV, FMDV, and EMCV 2B proteins did not inhibit the protein transport [36,46,76]. The FMDV 2B and 2C proteins did not block protein secretion, whereas the transport of proteins from the ER to the Golgi complex were blocked by the FMDV 2BC protein, and this effect was reproduced upon co-expression of the *2C* and *2B* genes [7,77]. Collectively, these findings suggest that the 2B protein may participate in a viral evasion of the host immune response, mainly by inhibiting protein transport. 

The 2B protein has also been suggested to facilitate the viral evasion of the host immune response through other means. Thus, the 2B protein antagonizes RIG-I-mediated antiviral responses by inhibiting the expression of RIG-I as an FMDV-specific reaction [14]. The RNA helicase LGP2 (also known as DExH-box helicase 58, DHX58) is a crucial factor involved in the host antiviral immune response [78]. The FMDV leader protein (Lpro), 3C protein, and the 2B protein have the ability to induce a decrease in LGP2 protein expression [79]. In addition, PV 2B variants were shown to inhibit the antiviral interferon (IFN) system [80], whereas the HAV 2B protein inhibited the synthesis of IFN-β by affecting the mitochondrial antiviral signaling protein activity, thereby antagonizing the host immune response [81]. Collectively, these evidences indicate that picornaviral 2B proteins can affect the host immune response, thereby promoting viral amplification or the release of viral particles. 

## 4. Potential Applications of Targeting the Picornaviral 2B Protein

As discussed above, picornaviral 2B proteins have a viroporin or viroporin-like activity and play an important role in the picornaviral life cycle. Therefore, many common applications targeting viroporins may be translatable to those targeting 2B proteins. In addition, the 2B protein may serve as a new target for the development of antiviral drugs. Thus, further studies on the structure and function of the 2B protein might open up new avenues for the prevention and control of picornaviruses. 

### 4.1. Detection of Picornaviruses

Owing to its highly conserved sequence, the use of the *2B* gene as a marker could effectively improve the accuracy of virus detection. Li et al. [15] designed primers and TaqMan probes, based on the *2B* and *3D* regions, which were successfully used in real-time polymerase chain reaction to accurately detect and quantify FMDV during infection and replication. In addition, Wang et al. [16] developed a lateral-flow detection system, which could rapidly and easily detect FMDV using the *2B* gene. In addition to gene-based detection, the virus could be detected using a 2B antibody. Biswal et al. [17] used an indirect enzyme-linked immunoassay based on a recombinant 2B protein to detect antibodies specific for FMDV. This method can be applied not only to FMDV but also to other picornaviruses, including CVB3 and EV71.

### 4.2. Development of Vaccines

Given the significant threat that picornaviruses pose to humans and animals, resulting in enormous economic damage to the livestock industry, development of picornavirus vaccines is of great significance. Although inactivated virus vaccines can offer effective prevention, there are associated residual risk issues, including incomplete virus inactivation and escape during the vaccine production process [82,83,84]. Therefore, genetically engineered vaccines are considered more suitable options to overcome these shortcomings of inactivated viral vaccines. The EV VP1 protein is located outside the viral membrane and is thus exposed to the greatest amount of immune stress. Accordingly, VP1 shows an extreme serological variability, thus providing the most reliable molecular epidemiological information. Consequently, the VP1 region of EV71 has become a focus of vaccine research for picornavirus infections [84]. However, DNA constructs containing the *VP1* gene of EV71 showed low levels of antigenicity. Therefore, there is still a need to develop an effective adjuvant strategy to increase the antigenicity. One possibility in this regard is the use of recombinant vaccines incorporating the *2B* gene to enhance the efficacy of vaccines.

At present, applications of the *2B* gene in recombinant vaccines have mainly concentrated on FMDV. The addition of a *2B* fragment to a vaccine designed with VP1 as the core has been shown to effectively enhance the vaccine efficacy [18,19,20] and reduce the dose and side effects [20]. These effects may be similar to those leading to a greater efficacy of the adenoviral vector vaccine fused to the FMDV 2B protein against serotype O, which is associated with the induction of specific CD4^+^ and CD8^+^ protective T cell responses [85]. Therefore, future designs of other picornaviral genetically engineered vaccines would benefit from the addition of the *2B* gene to increase the vaccine efficacy, including the addition of the *2B* gene to a genetically engineered EV vaccine with the *VP1* gene as the core.

### 4.3. Drug Development Strategy for Picornavirus Treatment

Since viroporin plays an important role in all life stages of the virus, it is an attractive antiviral therapeutic target, and there have been great breakthroughs in this regard. By contrast, research and development of drugs targeting the 2B protein are relatively delayed. Since 2B proteins have a viroporin or viroporin-like activity, screening for anti-picornavirus drugs among existing viroporin-targeting drugs may be a viable approach.

There are four main types of inhibitors of viroporin activity, including adamantane, amiloride, alkyl iminosugar, and spirane amine [86]. Adamantane (amantadine and rimantadine) inhibits the M2 channel of influenza A virus by destroying the transmembrane network of hydrogen-bonded water molecules, thereby inhibiting the viral amplification [87]. In BHK-21 cells infected with FMDV, the virus titer gradually decreased with an increase in amantadine concentration, which may have been due to abrogation of the pore-forming activity of the 2B protein and ultimate inhibition of FMDV replication [28]. However, clinical trials showed that amantadine was not only selective for specific resistance mutations in hepatitis C virus (HCV) p7 [88], but also caused a rapid emergence of amantadine-resistant variants of influenza A virus during monotherapy for influenza [89]. Amiloride is a composite of two drugs, 5-(*N*,*N*-hexamethylene) amiloride and a novel inhibitor, BIT225, targeting HCV p7 and HIV-1 Vpu, which can together block the viroporin ion channel activity or prevent ion channel formation, resulting in a potent antiviral effect [90,91,92,93]. The alkyl iminosugar inhibits the formation of ion channels by targeting the HCV p7 viroporin [94]. Finally, spirane amines, such as BL-1743, also inhibit the influenza A virus M2 protein, with an antiviral mechanism similar to that of amantadine [95]. 

There are also other drugs that act as viroporin inhibitors, including 1,3-dibenzyl-5(2*H*-1,2,3,4-tetraazol-5-yl) hexahydropyrimidine (CD), *N*-(1-phenylethyl)-2-[4-(phenylsulfonyl)-1-piperazinyl]-4-quinazolinamine (LDS25), and 6-methyl-1,3,8-trihydroxyanth-raquinone (Emodin), among others [88,96,97]. The mechanism of action of these viroporin inhibitors is based on the inhibition of the viroporin channel activity. Therefore, these drugs may have the potential to be applied for the treatment of picornavirus infections by targeting the *2B* gene. However, this application will require further detailed investigations and drug screening. Nevertheless, the 2B protein has the potential to widen the range of antiviral treatment strategies.

Furthermore, specific degradation of complementary mRNA can be triggered by small interfering RNAs (siRNAs) or folded short hairpin RNAs (shRNAs) [98], which can be explored as an RNA interference strategy, a relatively novel technology that has already been applied to treat many important pathogens, including HIV-1, hepatitis B virus, and herpes simplex virus [99,100,101]. Currently, shRNAs targeting the highly conserved *2B* gene sequence are widely used in picornavirus research, including FMDV [21,25], EMCV [23], and CVB3 [24], and significant experimental viral suppression has been achieved. Basically, RNA interference against *2B* gene affects the stability and integrity of the whole viral genome. The high nucleotide sequence conservation makes the *2B* gene an attractive target for RNA interference, which may potentially be effective against multiple picornavirus types, and open the door for additional siRNA drugs.

To date, there have been few studies specifically focusing on inhibitors of the 2B protein. Xie et al. [102] found that 4,4′-diisothiocyano-2,2′-stilbenedisulfonic acid (DIDS) blocked a chloride-dependent current, mediated by the EV71 2B protein, and suppressed viral amplification. However, further research is needed to uncover the underlying mechanism. Despite the many challenges in drug development, new technologies such as Fourier-transform infrared spectroscopy and design of molecular dynamics analogs, as well as cryo-electron microscopy and spectroscopy, are expected to greatly contribute to the development of antiviral drugs. 

## 5. Future Perspectives and Conclusion

Recent studies have gradually clarified the function and the potential of the 2B protein, along with increasingly recognizing its importance in the viral life cycle. However, there are still some challenges to overcome in investigations of the picornaviral 2B protein. In particular, its strong hydrophobicity makes it difficult to achieve soluble expression. Ao et al. [28] conjugated the small ubiquitin-like modifier (SUMO) protein to the N-terminus of the FMDV 2B protein and successfully achieved soluble expression. Therefore, this method can be tested for other picornaviral 2B proteins. Moreover, the detailed molecular mechanism of the action of the 2B protein requires further study, along with the identification of interactions of 2B protein with host proteins, to better understand the role of the 2B protein in the pathogenesis of picornaviruses. In murine cells, the 2B protein was suggested to react with host proteins to promote rhinovirus proliferation [103]. Using a yeast two-hybrid system, the FMDV 2B protein was found to interact with the host elongation factor 1γ (EEF1G), and mislocalization of EEF1G demonstrated that the EEF1G deletion affected the synthesis of membrane proteins [104,105]. Although a yeast two-hybrid system is a common laboratory protein-screening technique, it has a low success rate and is time-consuming. Alternatively, affinity purification–mass spectrometry can be used to overcome these shortcomings, which has already been widely used in studies on Dengue, Zika, and Ebola viruses [106,107]. 

At present, the development of antiviral drugs against viroporins is focused on three aspects, including viroporin and membrane fusion inhibitors, ion channel inhibitors, and targeted viroporin antibodies [9]. With respect to the biological function of the 2B protein, antiviral drugs targeting the 2B protein could be designed based on the following three approaches: broad-spectrum screening for anti-picornavirus drugs among existing viroporin inhibitors, screening for 2B protein and membrane fusion inhibitors, and screening for 2B protein pore activity inhibitors. As discussed herein, the most important basis for the function of the 2B protein is that it can be polymerized into pores, thereby changing the permeability of the membrane. Therefore, the design of drugs targeting 2B protein should be based on inhibiting polymerization of the 2B protein into pores, thereby reducing its effects on cellular ion homeostasis. However, these designs first require detailed determination of the refined atomic structure of the 2B protein, along with the expansion of screening techniques and applications of meticulous medicinal chemistry. 

Furthermore, to develop better antiviral drugs, it will be necessary to elucidate the exact role of the 2B protein channel in the viral life cycle. Thus, the main points of focus for research on the structure and function of the 2B protein toward ultimate drug development are: (1) mechanism of increasing membrane permeability to disturb the ion balance, (2) regulation of autophagy and apoptosis, (3) inhibition of the host immune response, and (4) promotion of viral replication and release. Taken together, as research aimed at further elucidation of the role of the 2B protein progresses, along with the adoption of new technologies, it is expected that more strategies will come to light for antiviral drug development and disease control.

## Figures and Tables

**Figure 1 viruses-11-00510-f001:**
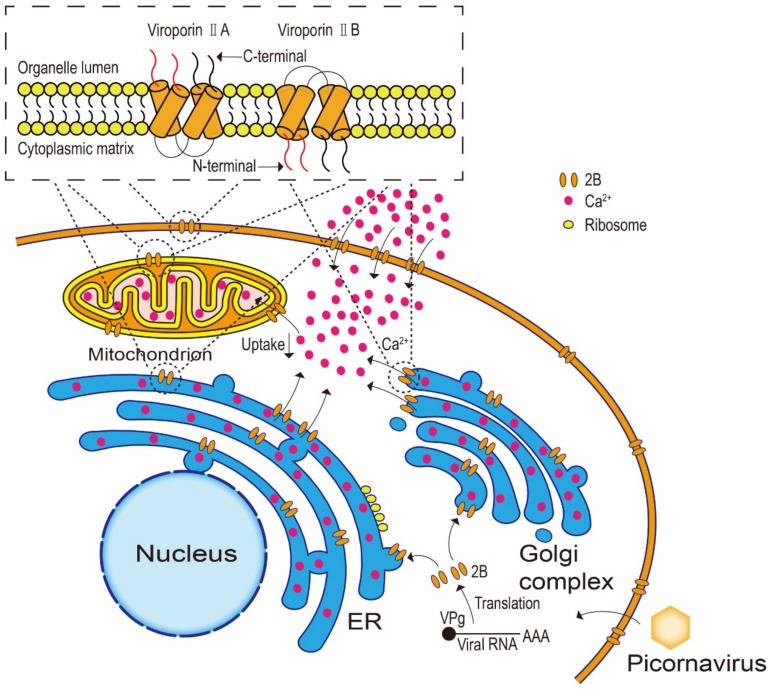
The structure and localization of 2B protein have effects on cell membrane permeability. Picornavirus infects the host cell, and then viral gene encodes 2B proteins. The 2B protein belongs to the type II family of viroporins, including two transmembrane hydrophobic helices which are the basis for forming a transmembrane pore that results in the changes in cell membrane permeability. In type IIA viroporins, the N- and C-termini stretch to the organelle lumen, whereas the N- and C-termini of type II B viroporins face the cytoplasmic matrix. The majority of 2B proteins are localized to organelles, with predominant co-localization with the Golgi apparatus and the endoplasmic reticulum (ER), resulting in an obvious decrease in Ca^2+^ of the ER and Golgi complex, along with a decrease in calcium uptake by the mitochondrion, and causing an influx of extracellular Ca^2+^.

**Figure 2 viruses-11-00510-f002:**
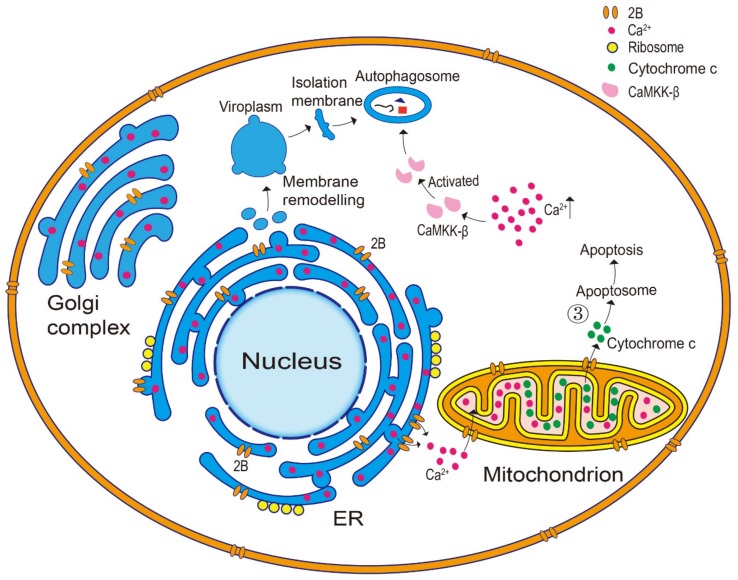
2B protein regulates autophagy and apoptosis. The 2B protein induces target membrane remodeling to form the viroplasm, which is generated from the endoplasmic reticulum (ER). The isolation membrane is produced by the viroplasm. Activation of the Ca^2+^/calmodulin-dependent kinase kinase-β (CaMKK-β) signal pathway is due to an increased intracellular calcium concentration. Furthermore, mitochondrion takes up Ca^2+^ from the ER, thereby cytochrome cis released, leading to apoptosis.

**Figure 3 viruses-11-00510-f003:**
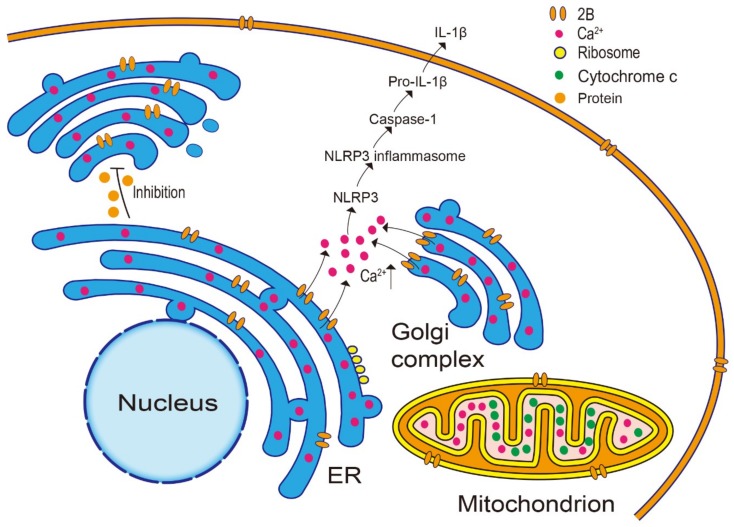
2B protein affects the host immune response. The 2B protein can stimulate the NLRP3 inflammasome pathway to activate caspase-1, which catalyzes the proteolysis of pro-IL-1β to IL-1β, which leads to their secretion across the plasma membrane by inducing a Ca^2+^ efflux from intracellular storage. Moreover, the 2B protein inhibits protein transport through the Golgi protein which may be effective to evade the host immune response.

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
