# Peer review of "Biological Function and Application of Picornaviral 2B Protein: A New Target for Antiviral Drug Development"

_viruses, 2019, doi:10.3390/v11060510_

Round 1
Reviewer 1 Report
In this manuscript, the authors attempt to review and summarize what is known about the biological function of picornaviral 2B proteins and their potential as targets for antiviral drugs.
Overall, the review is well organized, however the English is poor and lacks flow, limiting sometimes the reader’s comprehension.
The word “viroporin” should be defined when mentioned for the first time.
The authors should cite the pioneer of Carrasco and/or Nieva, perhaps the first scientists reporting the existence of these proteins in the 70’s. Not only these authors have contributed original discoveries on the topic but they have written great reviews, published in top journals (which by the way are not mentioned in the manuscript).
Section 2 on structure seems too abbreviated and is not always correct. I would encourage the authors to try to improve figure 1 or add an extra figure summarizing the multiple structures 2B proteins adopt including: oligomerization and orientation, Ca influx/efflux directionality for each organelle, and biological relevance of organelle specific Ca concentration change. The legend is oversimplified and hard to follow. Perhaps the authors should emphasize that Ca is needed for virus morphogenesis, entry and/or release.
The RNAi section may be deleted, or perhaps included as a short paragraph in 4.4. Basically RNAi of 2B affects the stability/integrity of the whole viral genome. High nucleotide sequence conservation makes it a nice target for RNAi with potential efficacy against multiple virus types.
Author Response
Reviewer 1
In this manuscript, the authors attempt to review and summarize what is known about the biological function of picornaviral 2B proteins and their potential as targets for antiviral drugs.
Overall, the review is well organized, however the English is poor and lacks flow, limiting sometimes the reader’s comprehension.
Our response:
Thank you very much for your review comments, we will fully consider your comments on this basis to further improve the manuscript. In the revised manuscript, we added the line number so that the reviewer could follow our revisions more easily. In addition, we have invited experts of American Journal Experts, which is a professional company for paper edition to edit this manuscript. The expert has read through this manuscript and edited the grammar and typing errors. We hope this revised manuscript looks concise and clear.
The word “viroporin” should be defined when mentioned for the first time.
Our response:
We thank the reviewer for this insightful comment. We have this sentence “Viroporin is generally composed of 50 to 120 amino acids and contains a highly hydrophobic domain capable of forming at least one amphipathic α-helical structure, which oligomerizes to form transmembrane hydrophilic pores” in lines 78-81 of revised manuscript.
The authors should cite the pioneer of Carrasco and/or Nieva, perhaps the first scientists reporting the existence of these proteins in the 70’s. Not only these authors have contributed original discoveries on the topic but they have written great reviews, published in top journals (which by the way are not mentioned in the manuscript).
Our response:
Carrasco and Nieva have made significant contributions to the study of Picornaviral 2B protein. So we quoted many of their articles and confirmed their work.
1. Viroporins are proteins found in a variety of viruses and are generally compriosed of 50 to 120 amino acids. Each Viroporins contains a highly hydrophobic domain capable of forming at least one amphipathic α-helical structure, which oligomerizes to form transmembrane hydrophilic pores. (line 81)
2. The 2B protein is a crucial component encoded by of picornaviruses that, exhibits viroporin or viroporin-like activity, playsing a key role in the picornavirus life cycle by inducing a series of cytotoxic reactions to promote picornaviral replication and release. (line 84)
3. In type IIA viroporins, the N- and C-termini stretch to the organelle lumen, such as in the 2B protein of CVB3,6, 9, 26, 33 whereas the N- and C-termini of type IIB viroporins face the cytoplasmic matrix, such as in PV (line 106)
4. A computational approach has demonstrated that the EV 2B protein is a tetramer, and 2B proteins with different orientations have different activities. (line 102)
5. The protein molecules are first inserted into the membrane individually and then self-interact and homooligomerize to form higher-order structures, which are important for the pore-forming activity, determined by the specific sequence and structure (line 113)
6. The changes in membrane permeability, caused by the 2B protein, have also been suggested to be regulated by the content of specific membrane phospholipids. (line 137)
7. The 2B protein has also been shown to regulate apoptosis through the endogenous pathway, ER stress, and under the mitochondrial pathway, providing another potential mechanism of by-passing the host immune response to facilitate infection. (line 196)
8. Excessive mitochondrial uptake of Ca2+ exerts a cytotoxic effect because a high Ca2+ concentration can open numerous mitochondrial transition pores, increase mitochondrial permeability, and destroy the mitochondrial outer membrane; consequently, cytochrome c and other proapoptotic factors are released, leading to apoptosis. (line 202)
9. Madan et al.1 showed that the PV 2B protein interacted with mitochondria and altered the mitochondrial morphology, in addition to the release of cytochrome c after PV 2B gene expression. (line 210)
Section 2 on structure seems too abbreviated and is not always correct. I would encourage the authors to try to improve figure 1 or add an extra figure summarizing the multiple structures 2B proteins adopt including: oligomerization and orientation, Ca influx/efflux directionality for each organelle, and biological relevance of organelle specific Ca concentration change. The legend is oversimplified and hard to follow. Perhaps the authors should emphasize that Ca is needed for virus morphogenesis, entry and/or release.
Our response:
Thank you very much for your review comments. To better explain the structure of the 2B protein, we have modified the description of Figure 1 legend to better characterize its structure. And the detail modification was showed in revised manuscript.
The RNAi section may be deleted, or perhaps included as a short paragraph in 4.4. Basically RNAi of 2B affects the stability/integrity of the whole viral genome. High nucleotide sequence conservation makes it a nice target for RNAi with potential efficacy against multiple virus types.
Our response:
Thank you for your comprehensive consideration, we have modified RNAi section based on your comments, and we have moved this section in the section of drug development strategy. The detail context is as below “Furthermore, specific degradation of complementary mRNA can be triggered by small interfering RNAs (siRNAs) or folded short hairpin RNAs (shRNAs),2 which can be explored as an RNA interference strategy, a relatively novel technology that has already been applied to treat many important pathogens, including HIV-1, hepatitis B virus, and herpes simplex virus.3-5 Currently, shRNAs targeting the highly conserved 2B sequence are widely used in picornavirus research, including FMDV,6, 7 EMCV,8 and CVB3,9 and significant experimental viral suppression has been achieved. Basically, RNA interference against 2B affects the stability and integrity of the whole viral genome. The high nucleotide sequence conservation makes the 2B gene an attractive target for RNA interference, which may potentially be effective against multiple picornavirus types, and open the door for additional siRNA drugs.” (lines 353-364)

Reviewer 2 Report
A comprehensive review is presented to cover the role of 2B protein in the infectivity cycle of the virus and its function. The figures are nicely designed and easy to understand.
Only a minor concern: last paragraph 3.3.. “The immune system is the primary line of defense…”. Would say that the primary line of defense is, raising the temperature in the body, until the immune system jumps in and presents the antibodies.
There should be some words being said about structural features of 2B to complete the overall good introduction to the 2B protein. There are some suggestions in the literature from experimental sources (J. Virology 1996, Virology 1997) and also some from computational modeling sources (Mol. Membr. Biol. 2009, Biochim Biophys Acta 2014).
Author Response
Reviewer 2
A comprehensive review is presented to cover the role of 2B protein in the infectivity cycle of the virus and its function. The figures are nicely designed and easy to understand.
Our response:
Thank you very much for your review comments, we will fully consider your comments on this basis to further improve the manuscript. In the revised manuscript, we added the line number so that the reviewer could follow our revisions more easily.
Only a minor concern: last paragraph 3.3.. “The immune system is the primary line of defense…”. Would say that the primary line of defense is, raising the temperature in the body, until the immune system jumps in and presents the antibodies.
Our response:
Thank you for your rigorous considerations. Due to our misrepresentation, you may not understand the original intent. We changed the original sentence to " The host immune system is an important line of defense against pathogens, and pathogens can affect the immune system in a variety of ways. " (lines 222-223)
There should be some words being said about structural features of 2B to complete the overall good introduction to the 2B protein. There are some suggestions in the literature from experimental sources (J. Virology 1996, Virology 1997) and also some from computational modeling sources (Mol. Membr. Biol. 2009, Biochim Biophys Acta 2014).
Our response:
Your comments are valuable. We have added more information about the structure of the 2B protein, and the detail context is as below “The α-helix–turn–α-helix sequence of the 2B protein is the basis for forming a transmembrane pore through homomultimerization and the major determinant of the 2B protein function.10-16 A computational approach has demonstrated that the EV 2B protein is a tetramer, and 2B proteins with different orientations have different activities” (lines 98-102)

Reviewer 3 Report
The review is informative. The abstract and introduction should be improved, mainly English.
Also, the first line of the abstract should be more clear. Picornaviruses are associated with acute and chronic diseases. Their clinical manifestations are often mild but their infections may lead to.. please improve the first line.
Author Response
Reviewer 3
The review is informative. The abstract and introduction should be improved, mainly English.
Our response:
Thank you very much for your review comments, we will fully consider your comments on this basis to further improve the manuscript. In the revised manuscript, we added the line number so that the reviewer could follow our revisions more easily. In addition, we have invited experts of American Journal Experts, which is a professional company for paper edition to edit this manuscript. The expert has read through this manuscript and edited the grammar and typing errors. We hope this revised manuscript looks concise and clear.
Also, the first line of the abstract should be more clear. Picornaviruses are associated with acute and chronic diseases. Their clinical manifestations are often mild but their infections may lead to.. please improve the first line.
Our response:
Thank you for your rigorous considerations and we have made improvements based on your comments. We have modified the abstract and replaced it by “Picornaviruses are associated with acute and chronic diseases. The clinical manifestations of infections are often mild, but infections may also lead to respiratory symptoms, gastroenteritis, myocarditis, meningitis, hepatitis, and poliomyelitis, with serious impacts on human health and economic losses in animal husbandry” in lines 24-27 of revised manuscript.
